# Translesion synthesis by AMV, HIV, and MMLVreverse transcriptases using RNA templates containing inosine, guanosine, and their 8-oxo-7,8-dihydropurine derivatives

**Madeline M. Glennon**[☯], **Austin Skinner**[iD][☯], **Mara Krutsinger**, **Marino J. E. Resendiz**[iD]*

Department of Chemistry, University of Colorado Denver, Denver, Colorado, United States of America

☯ These authors contributed equally to this work.
* marino.resendiz@ucdenver.edu

**Data Availability Statement:** All relevant data are within the manuscript and its Supporting Information files.

## Abstract

Inosine is ubiquitous and essential in many biological processes, including RNA-editing. In addition, oxidative stress on RNA has been a topic of increasing interest due, in part, to its potential role in the development/progression of disease. In this work we probed the ability of three reverse transcriptases (RTs) to catalyze the synthesis of cDNA in the presence of RNA templates containing inosine (I), 8-oxo-7,8-dihydroinosine (8oxo-I), guanosine (G), or 8-oxo-7,8-dihydroguanosine (8-oxoG), and explored the impact that these purine derivatives have as a function of position. To this end, we used 29-mers of RNA (as template) containing the modifications at position-18 and reverse transcribed DNA using 17-mers, 18-mers, or 19-mers (as primers). Generally reactivity of the viral RTs, AMV / HIV / MMLV, towards cDNA synthesis was similar for templates containing G or I as well as for those with 8-oxoG or 8-oxoI. Notable differences are: 1) the use of 18-mers of DNA (to explore cDNA synthesis past the lesion/modification) led to inhibition of DNA elongation in cases where a G:dA wobble pair was present, while the presence of I, 8-oxoI, or 8-oxoG led to full synthesis of the corresponding cDNA, with the latter two displaying a more efficient process; 2) HIV RT is more sensitive to modified base pairs in the vicinity of cDNA synthesis; and 3) the presence of a modification two positions away from transcription initiation has an adverse impact on the overall process. Steady-state kinetics were established using AMV RT to determine substrate specificities towards canonical dNTPs (N = G, C, T, A). Overall we found evidence that RNA templates containing inosine are likely to incorporate dC > dT > > dA, where reactivity in the presence of dA was found to be pH dependent (process abolished at pH 7.3); and that the absence of the C2-exocyclic amine, as displayed with templates containing 8-oxoI, leads to increased selectivity towards incorporation of dA over dC. The data will be useful in assessing the impact that the presence of inosine and/or oxidatively generated lesions have on viral processes and adds to previous reports where I codes exclusively like G. Similar results were obtained upon comparison of AMV and MMLV RTs.

**Funding:** This work was supported, in part, via an Office of Research Services grant from the University of Colorado Denver. Funding from NIGMS (1R15GM132816) is also acknowledged. Characterization of oligonucleotides was carried out at the University of Colorado's Bruker Center for Excellence (Department of Pharmaceutical Sciences, Skaggs School of Pharmacy and Pharmaceutical Sciences, The University of Colorado Anschutz Medical Campus), partially funded by the L.S. Skaggs Professorship and NIH grant R35GM128690, under the guidance of Mr. Justin Jens (laboratory of Prof. Vanessa V. Phelan); or at the Proteomics and Metabolomics Facility at Colorado State University. Funds from NSF-MRI-1726947 facilitated this work.

**Competing interests:** The authors have declared that no competing interests exist.

## Introduction

There are two aspects that provided the motivation for this work, one regards to the importance of inosine (naturally occurring modification), and the other to the impact that oxidatively generated lesions (chemically formed from endogenous or exogenous sources) within RNA have on enzymatic processes, reverse transcription (RTn) in this case. Another aspect of interest was to gain an understanding of how different modifications or lesions behave in various contexts, which increases their potential use as tools to unravel mechanistic aspects of biologically relevant pathways, e.g., the use of 8-bromopurines to explore conformational & H-bonding changes; and the use of 8-oxoinosine or inosine to explore the role of the C2-exocyclic amine in 8-oxoG or G (modifications used in this work).

Inosine (I) is the deamination product of adenosine (A) (Fig 1) and its formation is catalyzed by deaminases that act on RNA (ADARs) [1,2]. Furthermore, IMP is the first nucleotide that is generated in the *de novo* purine synthesis pathway and is then enzymatically derivatized to yield the corresponding AMP or GMP (via XMP) [3]. Examples that highlight the importance of this modification include that: 1) it is commonly observed in a variety of functions that include editing (changes from A to I alter the H-bonding interactions and, as a consequence the coding properties of mRNA), e.g., in the maturation of tRNA [4,5]; 2) it has also been identified in short RNAs such as micro-RNAs, albeit at lower levels than their longer precursors (pri-miRNA) [6]; and 3) its presence can cause ribosome stalling [7]. Furthermore, this modification has been associated with disease, along with xanthosine and 8-oxoG [8], and various human pathologies (e.g., as profiled in the inosinome atlas [9]). It is no surprise then of the existence of enzymes that specifically cleave RNA containing inosine [10], or that remove them from cellular nucleotide pools [11]. It is also important to note that, while the presence of I has been characterized as a marker of viral infection, using respiratory syncytial virus-RSV as model [12]; its incidence on other viral and cellular RNAs has been quantified at low levels on cells infected with Zika virus, Dengue virus, hepatitis C virus (HCV), poliovirus and human immunodeficiency virus type 1 [13]. Thus bringing into question the relevance of this modification in distinct viral RNAs. However, we hypothesize that it is plausible that reverse transcriptases encounter inosine within viral or cellular RNA, thus understanding how they cope with its presence within templates of RNA is of importance to understand potential

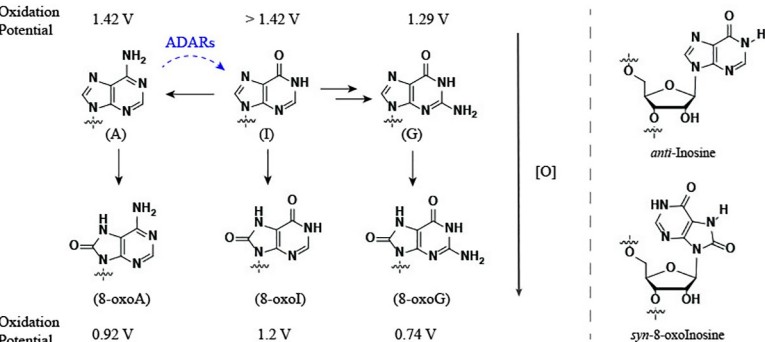

**Fig 1. Diagram showing structures of the nucleobases of interest.** Horizontal black arrows denote the role of inosine in the *de novo* purine synthesis of nucleotides (two arrows between I and G represent intermediate xanthosine in the process). Dashed blue curved arrow represents the formation of I from A, catalyzed by adenosine deaminases acting on RNAs. Vertical arrows represent the structure of one of the oxidation products of each purine at the C8-position (left). Purine nucleotides can undergo an anti→syn conformational change upon C8-modification (right). The values shown for the oxidation potentials were taken from literature and the experimental setup is not the same for all [ref. [26] for G/A; [27] for I; and [28] for 8-oxoA/8-oxoI/8-oxoG].

mechanisms, outcomes and/or strategies addressing the synthesis of viral DNA or other factors involving reverse transcription.

On a different note, oxidative damage of RNA is a topic that has increasingly captured attention due to its potential role in the development/progression of disease [14–17]. Oxidized RNA has been shown to occur in various types of RNA including rRNA [18], miRNA [19], or mRNA [20], and intracellular mechanisms in charge of diminishing the impact of oxidation have been reported [21–24]. Furthermore, the role of oxidative stress on viral pathogenesis [25] is a factor that could lead to interactions between reverse transcriptases and oxidatively damaged RNA. The trend for the oxidation potential of purine nucleobases (I > A > G) [26,27] makes G the likeliest candidate to undergo transformation under oxidative stress. In fact, 8-oxoG is one of the most abundant oxidatively generated lesions, a result of oxidation at the C8-position, with trends in their redox properties matching those of the precursor structures (8-oxoI > 8-oxoA > 8-oxoG) [28]. Although this physical property decreases the probability that 8-oxoI or 8-oxoA are formed, rendering them not as biologically relevant as 8-oxoG, the lesions have attracted interest in other contexts. For example in their potential role in a prebiotic scenario, where it was established that none of the oxidized versions of G, A, or I (8-oxoG, 8-oxoA, 8-oxoI) are likely substrates for prebiotic RNA replication. Interestingly, inosine was found to be a possible candidate in this regard [29]. Other examples highlighting their use as tools to probe for biochemical mechanisms include, where 8-oxoinosine and other C8-subsituted purines have been employed to understand the base excision of 8-oxoG by the MutY glycosylases [30]; or in the replication of DNA with *E. coli* DNA polymerase I [31].

Therefore, considering the prominent role that inosine has on various biological processes and that oxidation of RNA is ubiquitous, with purines undergoing oxidation to their corresponding 8-oxo-7,8-dihydropurine derivatives, we decided to probe these modifications/lesions within RNA templates using reverse transtrition (RTn) as framework. Briefly, RTn begins by association of a reverse transcriptase (RT) with a nucleic acid substrate, typically composed of a primer bound to a complementary template with the polymerase domain of RT interacting with the primer 3' terminus which is recessed on the template. This is followed by formation of a ternary complex with the incoming dNTP to form a phosphodiester bond; and nucleic acid translocation relative to the RT to continue with processive DNA synthesis [32]. Structural differences among all RTs used in this work are well established and have been reviewed for HIV RT [33], AMV RT [34], and MMLV RT [35]. Notably both AMV and MMLV RT have a sequence homology of 23% and contain the same five domains needed for RTn [36], while the HIV RT has structural differences that lead to distinct fidelity *in vitro*.

Previous work on reverse transcription of RNA containing inosine has been reported using an engineered form of MMLV RT, Superscript IV, to show that I codes like G in experiments aimed at developing deep sequencing technologies [37]. Furthermore, reverse transcription of RNA containing 8-oxoG in the presence of AMV or MMLV RTs showed insertion of A and C opposite this lesion [38]. In this work, we were interested in: 1) probing the impact of the exocyclic amine on both G and 8-oxoG, thus we directly compared strands of RNA containing this nucleobases to RNA containing I or 8-oxoI; 2) extending the impact of reverse transcription to HIV RT; and 3) establishing the reactivity of the reverse transcriptase within the vicinity of the reactive site containing G, I, 8-oxoG, or 8-oxoI (for which three DNA primers of varying length were used).

Interestingly some unexpected reactivity was observed on RNA containing I, where experiments carried out in the presence of dTTP or dATP allowed for the enzyme to facilitate their incorporation into DNA opposite this modification. Steady-state kinetics showed that the order of selectivity was dC > dT >> dA and that there was a dependence on pH, where decreasing the pH led to a lack of recognition towards dA and decreased recognition towards

dT. RNA templates containing 8-oxoG displayed the same reactivity as previously reported, where either dC or dA are incorporated. Furthermore, reverse transcription in the presence of RNA templates containing 8-oxoI or 8-oxoG displayed a higher efficiency towards the incorporation of dA, compared to that observed for dC, for the former and suggests that the exocyclic amine plays a role in the fidelity and efficiency of the enzyme. Lastly, besides the potential biological relevance, it is important to establish the reactivity of reverse transcriptases in vitro, where sequencing and the effects of various modifications was highlighted recently [39]. It is useful to understand their specificity/reactivity in the presence of various modifications that are potentially found/generated in biological processes.

## Materials and methods

### General

The detailed synthesis for the phosphoramidites of 8-oxoI and 8-BrI is included in the (S1 and S2 Files). It includes spectroscopic information for all known and novel intermediates. $^1$H NMR, $^{13}$C NMR, and $^{31}$P NMR spectra, recorded at 300, 75, and 121.5 MHz respectively (using a standard broadband multinuclear probe on a 300MHz Avance III platform from Bruker); IR spectra were recorded on a diamond ATR sampler using powders of pure materials; high-resolution mass spectrometry was carried out via ESI/APCI; and UV-vis spectroscopy of all small molecules was carried out on a Perkin Elmer λ-650 UV/vis spectrometer using quartz cuvettes (1 cm pathlength). The synthesis corresponding to the phosphoramidite for 8-oxoG was carried out according to a previous report [40]. All experiments described herein were carried out in triplicate, unless otherwise noted. The protocol for the reverse transcription experiments is available at: dx.doi.org/10.17504/protocols.io.birskd6e.

### RNA synthesis

Oligonucleotides were synthesized on a 394 ABI DNA/RNA synthesizer using CPG supports and 2'-O-TBDMS phosphoramidites (purchased from Glen Research). 0.25M 5-Ethylthio-1H-tetrazole in acetonitrile was used as the coupling reagent; 3% trichloroacetic acid in dichloromethane was used for deblocking; a 2,6-lutidine/acetic anhydride solution was used for capping; and an iodine (0.02 M) in/THF/pyridine/water solution was used in the oxidation step (also purchased from Glen Research). Coupling times of 10 min were used. Oligonucleotides (ONs) were deacetylated / debenzoylated / deformylated and cleaved from the CPG support in the presence of 1:1 aq. methylamine (40%) and aq. ammonia (40%) with heat (60° C, 1.5 h). A mixture of *N*-methylpyrrolidinone/triethylamine/HF (3:2:1) was used for deprotection of the TBDMS groups (60° C, 1 h) followed by purification via electrophoresis (20% denaturing PAGE). C18-Sep-Pak cartridges were obtained from Waters and used to desalt the purified oligomers using 5 mM NH$_4$OAc as the elution buffer. Oligonucleotides were dissolved in H$_2$O and used as obtained for subsequent experiments. Unmodified ONs were purchased from IDT-DNA or ChemGenes and, following quantification via UV-vis, used without further purification.

### RNA characterization (MALDI-TOF)

Mass spectra (MALDI-TOF MS) for all the modified oligonucleotides were obtained using C18 Zip Tip pipette tips to desalt and spot each ON as follows: 1) wash tip with 50% acetonitrile (10 μL x 2); 2) equilibrate tip with 0.1% TFA (10 μL x 2); 3) load tip with sample (typically 100–150 picomol); 4) wash tip with 0.1% TFA (10 μL x 2); 5) wash tip with water (10 μL x 2); 6) elute sample into matrix (10uL of 25 mM-2,4,6-trihydroxyacetophenone monohydrate, 10mM ammonium citrate, 300 mM ammonium fluoride in 50% acetonitrile); 7) spot directly

onto MALDI plate. All analyses were carried out on an ABI 4800 Plus MALDI-TOF/TOF mass spectrometer in positive mode (see acknowledgments and experimental section for further details, S3 File).

## UV-vis spectroscopy

Concentrations of all oligonucleotides (no secondary structure was detected, via CD, for any oligonucleotide used/measured herein) were obtained via UV-vis using a 1 mm path-length with 1 μL volumes (Thermo Scientific Nano Drop Nd-1000 UV-vis spectrometer). Origin 9.1 was used to plot the spectra of monomers and oligonucleotides for comparison. UV-vis spectra for the reverse transcriptases used in this work were obtained using a 1 mm path-length of neat samples from commercial RTs AMV and MMLV, or a 10-fold diluted (1 μL enzyme in 9 μL of water) sample of the corresponding HIV RT (S4 File).

## Circular dichroism (CD) spectroscopy and thermal denaturation transitions ($T_m$)

CD spectra were recorded at various temperatures (PTC-348W1 peltier thermostat) using Quartz cuvettes with a 1 cm path length. Spectra were averaged over three scans (325–200 nm, 0.5 nm intervals, 1 nm bandwidth, 1 s response time) and background corrected with the appropriate buffer or solvent. Solutions containing the RNA strands had the following composition: 1.5 μM RNA, 2 μM DNA, 5 mM MgCl2, 10 mM NaCl, 1 mM sodium phosphate-pH 7.3. All solutions used to record thermal denaturation transitions ($T_m$) were hybridized prior to recording spectra by heating to 90˚C followed by slow cooling to room temperature. $T_m$ values were recorded at 270 nm with a ramp of 1˚ / min and step size of 0.2 with temperature ranges from 4˚C to 95˚C. A thin layer of mineral oil was added on top of each solution to keep concentrations constant at higher temperatures. Origin 9.1 was used to determine all $T_m$ values and to normalize CD spectra of ss-RNA and ds-oligonucleotides for all RNA:DNA duplexes. Analysis of variance (ANOVA) was carried out to determine significance amongst groups of values (S5 File).

## Oligonucleotide radiolabeling

T4 polynucleotide kinase (PNK) and γ-$^{32}$P-ATP-5′-triphosphate were obtained from Perkin Elmer. Oligonucleotides were labeled by mixing PNK, PNK buffer, ATP, DNA, and water (final volume = 50 μL) according to manufacturer's procedure followed by incubation at 37˚ C for 45 min. Radiolabeled materials were passed through a G-25 sephadex column followed by purification via electrophoresis (20% denaturing PAGE). The bands of interest (slowest migrating) were extruded, crushed, and soaked in a saline buffer solution (0.1 M NaCl) for 12 h at 37˚ C. The remaining solution was filtered and concentrated to dryness under reduced pressure followed by precipitation in a NaOAc (300 mM) / ethanol 1:5 solution (by volume) using a dry ice/ethanol bath. Supernatant was removed and the remaining oligonucleotide was concentrated under reduced pressure and dissolved in water. Activity was assessed using a Beckmann LS 6500 scintillation counter. Electrophoresis was not necessary for DNA strands previously purified via HPLC (purchased from manufacturer). A tracking dye (90% formamide containing 0.05% xylene cyanol and 0.05% bromophenol blue) was added to a well on each side of the gel; and this procedure was repeated for all subsequent experiments.

## Electrophoretic mobility shift assays

Radiolabeled oligonucleotides were mixed in buffers under the desired conditions and all samples were heated to 90˚ C with slow cooling to room temperature before loading. All samples

were electrophoresed using 20% non-denaturing PAGE (10 × 8 cm). Samples were typically mixed in a 1:1 mixture with 75% glycerol loading buffer. Quantification of radiolabeled oligonucleotides was carried out using a Molecular Dynamics Phosphorimager 840 equipped with ImageQuant Version 5.1 software.

### Reverse transcription experiments

Four reverse transcriptases were used in this work, Moloney Murine Leukemia Virus (MMLV), SuperScript II (genetically engineered MMLV reverse transcriptase (RT) with reduced RNase H activity and increased thermal stability), Avian Myeloblastosis Virus (AMV), and Human Immunodeficiency Virus-1 (HIV-1). They were obtained from commercial sources and details are delineated in Table 1. Reactions were carried out in the buffers suggested and provided by the manufacturer as follows: MMLV RT—5 mM Tris-HCl, 7.5 mM KCl, 0.3 mM $MgCl_2$, 1 mM DTT (pH 8.3); AMV RT—5 mM Tris-acetate, 7.5 mM Potassium Acetate, 0.8 mM Magnesium Acetate, 1 mM DTT (pH 8.3); SSII—25 mM Tris-HCl, 37.5 mM KCl, 1.5 mM $MgCl_2$, 10 mM DTT (pH 8.3). Reactions corresponding to the HIV RT were carried out using the buffer for AMV RT. dNTPs (dGTP, dCTP, dATP, dTTP) were purchased from Thermo Fisher or New England Biolabs (at a concentration of 100 mM) and diluted to a final concentration of 1.7 or 0.5 mM per experiment (single tube and well loaded onto the gel). dNTP mix, typically labeled M on subsequent figures, was diluted to a final concentration of 425 μM in each dNTP per experiment. It is important to note that enzyme concentrations are lower than those recommended by the manufacturer, thus the concentration of RNA:DNA duplex or dNTP must not be compared to those conditions. Decreasing the dNTP concentration to recommended values (350–500 μM) led to lower efficiencies with the same addition trends. Additionally, experiments where the [dNTP] was higher than 15–20 mM resulted in no reaction overall (no bands corresponding to dNTP incorporation were detected). It is important to note that the Mg ion concentrations varied with each of the recommended conditions, and the amount of free $Mg^{2+}$ was calculated as previously reported [41], to yield values of: 10, 312, or 371 μM for AMV and HIV RTs; 6, 12.2, or 15.8 μM for MMLV RT; and 0.45, 1, or 1.1 mM for SSII RT. Where values correspond to dNTP concentrations of 1.7, 0.5, or 0.425 mM respectively.

Fresh solutions were prepared for each set of experiments. Typical experiments for reverse transcription were carried out by preparing solutions containing the DNA primer of interest

**Table 1. Details for the reverse transcriptases used in this work.**

| Entry | RT | Source | Units/μL | Units [low]* | Units [High]* | Concentration† | RT amount—low (pg)§ | RT amount—high (pg)§ | RT UV-vis‡ |
|---|---|---|---|---|---|---|---|---|---|
| 1 | AMV | NEB | 10 | 0.7 | 2.1 | 0.036 mg/mL | 2,520 | 7,560 | nd |
| 2 | MMLV | NEB | 200 | 0.03 | 0.07 | 2 mg/mL | 300 | 690 | nd |
| 3 | HIV | Worthington | 27–28 | 0.006 | 0.19 | 1.81 mg/mL | 398 | 12,576 | 0.017 pmol |
| 4 | SS-II | Thermo Fisher | 200 | 4 | – | – | – | – | – |

* These columns represent the number of units used per well (single experiment). Unit definition: HIV -One Unit incorporates 1 nmole of tritiated d-TMP into acid precipitable products using poly(A)/oligo(dT)12-18 as the template/primer in 20 minutes at 37°C, pH 8.3; MMLV or AMV—One unit incorporates 1 nmol of dTTP into acid-insoluble material in a total reaction volume of 50 μl in 10 minutes at 37°C using poly(rA).oligo(dT) as template primer with 50 mM Tris-HCl (pH 8.3), 6 mM $MgCl_2$, 10 mM dithiothreitol, 0.5 mM [3H]-dTTP and 0.4 mM poly(rA).oligo(dT)12-18.

† This value represents the concentration of the stock vial provided by manufacturer (different batches of enzyme varied slightly).

§ This value represents the number of grams used for both low and high concentrations used in this work, per well/experiment.

‡ This value was obtained from UV-vis measurements of each RT using reported extinction coefficients (ε values for HIV RT is known [42] while AMV and MMLV values have not been reported).

(radiolabeled at the 5'-end at approx. 4,000 counts per electrophoresis well), the RNA template of interest in 2–5 fold molar excess with respect to the DNA (from prepared 1 or 5 μM working solutions), buffer provided by manufacturer (10x provided by manufacturer and diluted to 1x to yield the concentrations described above), and water to a final volume of 20 μL. For example, a cocktail solution was prepared as follows: 5'-$^{32}$P*-DNA (1 μL, < 1 pmol), RNA (2 μL, 2pmol), 10x buffer (2 μL), and water (15 μL); and annealed by placing in a heat block at 90˚C followed by slow cooling to room temperature (over app. 1–1.5 h).

**Reverse transcription (RTn).** The annealed solution (3 μL) was transferred to a new tube followed by addition of dNTP (or water for control experiments, 1 μL), and RT (2 μL). The mixture was then incubated at 37˚C (or room temperature) for the desired amount of time. Addition of loading buffer (6 M urea) followed, along with mixing and heating to 90˚C for 5–10 min. The tube(s) were then allowed to cool down to rt and centrifuged before loading onto a 20% denaturing PAGE (43 × 35 cm). A voltage was applied to the gels until the xylene cyanol dye passed ¾ the length of the gel (dyes were added on both ends of the gel), followed by exposure using an autoradiography cassette (Amersham Biosciences) overnight. Quantification of radiolabeled oligonucleotides was carried out using a Molecular Dynamics Phosphorimager 840 equipped with ImageQuant Version 5.1 software.

**Steady-state kinetics.** A solution containing the duplex of interest, in the corresponding buffer, was placed on a heat block at 90˚C followed by slow cooling to room temperature (over app. 1–1.5 h). A solution containing the dNTP at concentrations that varied with the nature of the RNA template (2 mM– 850 nM) was mixed with the anneal solution, followed by addition of the reverse transcriptase. These reaction mixtures were then incubated at 37˚C for 5, 10, 15, or 45 minutes depending on the process being measured. The mixtures were quenched with loading buffer (6M urea), followed by heating of the sample at 90˚C for 5–10 minutes. The samples, each containing various dilutions of dNTP, were loaded on to a 20% denaturing PAGE, and developed as described above. Kinetics ($V_{max}$, $K_m$, and $V_{max}/K_m$) were determined using a Hanes-Woolf plot, and assessed using multiple experiments per graph (3–5 gels/plot). Reaction velocities were measured at various substrate concentrations and the values for $V_{max}$ and $K_m$ were obtained by plotting [S]/V vs [S] to obtain a linear graphical representation where: slope = $1/V_{max}$; $K_m$ = y-intercept•$V_{max}$; $K_{cat}$ was calculated by dividing $V_{max}$ over 0.7 (units used per experiment). Excell was used in all calculations. Different enzyme batches, from same manufacturer, were used throughout.

**Single-point kinetics—relative rates at constant [dNTP] and [RT] as a function of time (see supporting info).** A solution containing the duplex of interest in the appropriate buffers was placed on a heat block at 90˚C followed by slow cooling to room temperature (over app. 1–1.5 h). A solution containing the reverse transcriptase was mixed into the reaction tube followed by addition of the dNTP. A determined volume was then withdrawn and added to a tube containing loading buffer (6M urea), followed by heating of the sample to 90˚C for 5–10 min. These steps were repeated at various time intervals followed by loading into a 20% PAGE, and developed as described before. Importantly for these experiments, the incubations were carried out at room temperature.

## Results

### Reverse transcription–cDNA synthesis

Reverse transcription was used to studycDNA synthesis with RNA (29-nt long) templates containing guanosine, inosine, 8-oxoinosine, or 8-oxoguanosine at position-18. The modified oligonucleotides were prepared using standard phosphoramidite chemistry via solid phase synthesis, purified, and characterized prior to their use. The selectivity and efficiency of each

RT for enabling the incorporation of canonical dNTPs was explored using DNA primers of three lengths (17-, 18-, or 19-nt long) to explore RTn: 1) opposite the modification/lesion; 2) past a modified/canonical base pair generated at the end; or 3) on an RNA:DNA duplex in which a modified base pair is present two positions away from the start of cDNA synthesis. The RTs used were Avian Myeloblastosis Virus (AMV); Moloney Murine Leukemia Virus (MMLV); human immunodeficiency virus (HIV); and a genetically modified version of MMLV (Superscript-II). The overall process was explored with canonical dNTPs (dG, dC, dT, dA) individually and as a mixture, to explore transcriptase selectivity and efficiency. In addition, the kinetic parameters of some reactions were derived to establish relative rates. We also compared the results to RNA templates containing 8-bromoinosine at the site of interest, which was used to probe for potential relationships between H-bonding and anti/syn conformational variations. This allowed us to establish potential differences among these lesions/modifications and gain a better understanding of their corresponding incorporation ratios. Since the only structural aspect differentiating I from G, and 8-oxoI from 8-oxoG, is the presence of an exocyclic amine at position 2 of the purine ring, we initially reasoned/expected that these pairs would exhibit similar outcomes. Experiments were carried out at two concentrations of RT while keeping the concentration of the corresponding dNTPs in excess (1.5–0.5 mM) with respect to the RNA:DNA duplexes (50–100 nM). In this manner, the effect of the RT was explored under higher RT concentrations that allowed us to establish reactivity and selectivity, and under lower RT concentrations that aided in identifying efficiency and specificity/selectivity (see Table 1).

**dNTP incorporation opposite the modification on duplexes 1:5–4:5.** To begin our studies, RNA templates (29-nt long) **1**–**4** were set for hybridization with the corresponding DNA primer, 17-mer (**5**) (Fig 2A). The sequence of the template was chosen based on a report by Alenko et al. [38], with the exception that the length of the RNA strand was extended by nine nucleotides to explore continuation of cDNA synthesis in more detail. All solutions were prepared in the buffers provided by the enzyme manufacturers (see experimental details). Formation of the corresponding duplexes was confirmed in two ways: 1) via CD using phosphate buffered solutions (10 mM sodium phosphate at pH 7.2, 1 mM NaCl, 5 mM MgCl$_2$, S5 File); and 2) via electrophoretic analyses (in buffer provided by manufacturer) where a slower band was observed on native PAGE gels, that can be assigned to hybridization of the corresponding RNA and DNA to their duplexes (S6 File). It is important to note that buffers recommended for RTn were not compatible with CD spectroscopy, given that strong absorption was observed at all wavelengths where features of an A-form duplex are expected (see S5 File for comparison between duplex and ss-RNA), thus the use of the phosphate buffered mixture mentioned above was employed for all experiments that required CD (including T$_m$ analyses). Thermal denaturation transitions corresponding to duplexes **1:5**–**4:5** led to values that were equivalent (Fig 2A—70°C). We initiated experiments using AMV RT (Fig 2B) to observe: 1) that the duplex where the template contains an I (**2:5**) enables the incorporation of dA while the analogous G-containing duplex (**1:5**) does not catalyze this process under the conditions described herein (lanes 15 and 5); 2) that the I-template exhibits lower incorporation selectivity between dC and dT than that observed on the G-template, which preferentially adds dC (Fig 2C); 3) that the G- or I- containing templates enabled the incorporation of dT, while the 8-oxopurine analogues do not (Lanes 4, 14, 9, 19); and 4) that duplexes containing 8-oxoG (**3:5**) or 8-oxoI (**4:5**) facilitated the incorporation of dA and dC with different efficiency (Fig 2C). Markedly, both G and I selectively incorporate pyrimidines dT and dC, while 8-oxoG and 8-oxoI selectively incorporate dC and dA (Lanes 3, 4, 13, 14, 8, 10, 18, 20).

Steady-state kinetics experiments were carried out by obtaining Hanes-Woolf plots for the processes of interest (Fig 2C) [43,44], which led to apparent $K_m$ values and $V_{max}$ as %

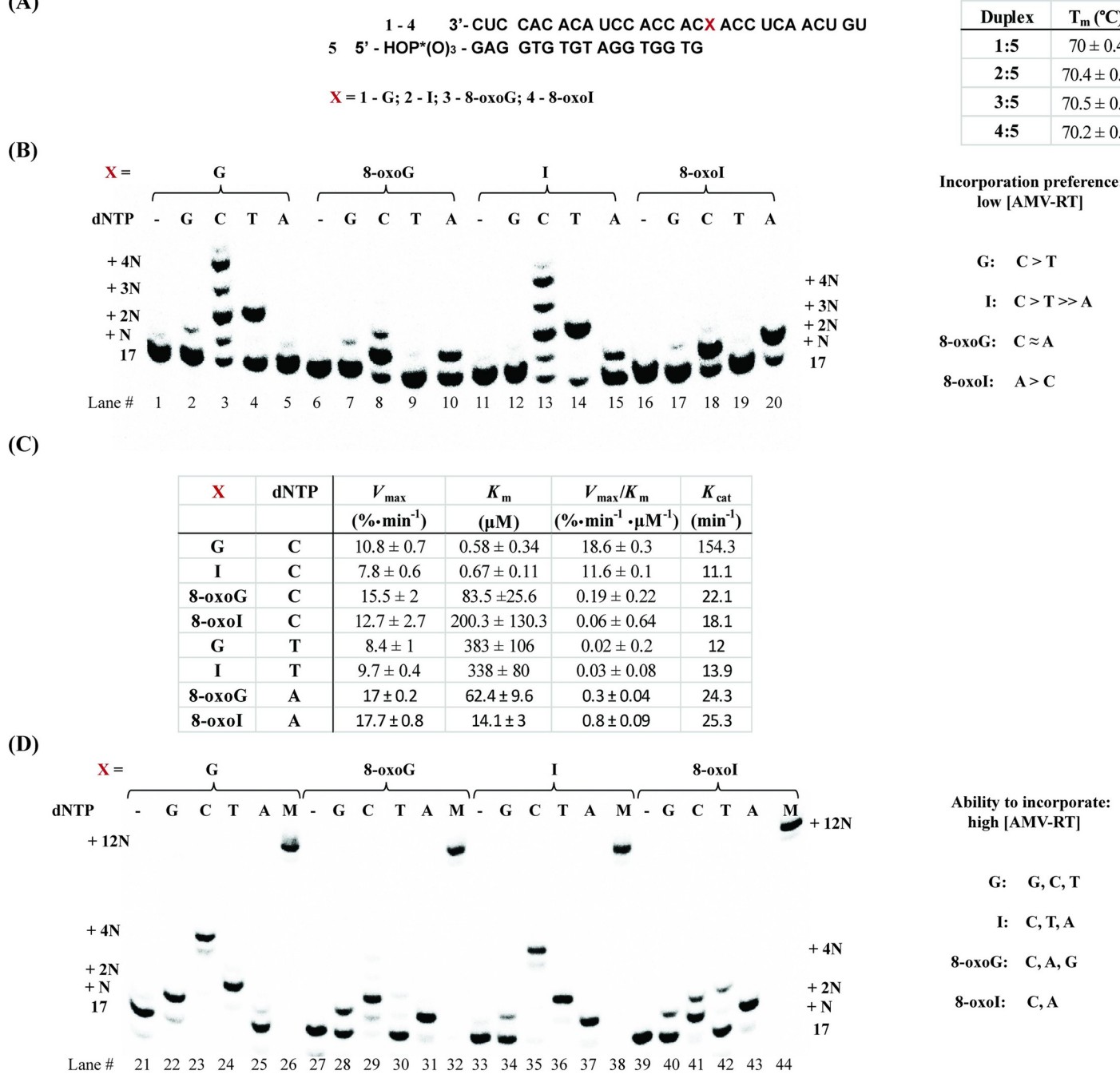

**Fig 2. RTn of duplexes 1:5–4:5.** (A) sequence and $T_m$ analysis of all duplexes in PBS buffer (10 mM sodium phosphate, 1 mM NaCl, 5 mM $MgCl_2$, pH 7.5), recorded via CD; (B) reverse transcription using AMV RT and duplexes **1:5–4:5** in the presence of low [AMV RT]; and (D) higher [AMV RT] under buffered conditions (5 mM Tris-acetate, 7.5 mM Potassium Acetate, 0.8 mM Magnesium Acetate (10 μM free $Mg^{2+}$), 1 mM DTT, pH 8.3, 37 °). (C) represents steady-state kinetics for the processes of interest.

conversion. The corresponding nucleotide insertion frequencies ($V_{max}/K_m$) were measured using the same enzyme concentration. Interestingly, incorporation of dNTPs followed the trends: dC for templates containing G > I >> 8-oxoG ≈ 8-oxoI; dA for templates containing 8-oxoI > 8-oxoG; and dT for templates containing G ≈ I. Additional experiments where side-

by-side reactions containing equal amounts of RNA:DNA duplex / RT / dNTP, while monitoring the formation of product as a function of time, yielded the same trend (S7 File). Of note is that templates containing 8-oxoI enable the incorporation of dA more efficiently than those containing 8-oxoG; while RNA templates containing 8-oxoG facilitate the incorporation of dC more efficiently than the corresponding 8-oxoI analogue. Furthermore, incorporation of dT in the presence of RNA templates containing G or I occurs about 3 orders of magnitude less efficiently than the analogous reaction using dCTP. Acquisition of kinetics data, using templates containing I, in the presence of dATP was not be obtained given the poor efficiency of the process (S7 File).

As shown in Fig 2D, reactions carried out at higher [AMV RT] led to the following differences: 1) there is addition of dG opposite G (lane 22), while the I-containing template does not enable the incorporation of dG (lane 34); 2) that higher enzyme concentrations lead to lower fidelity on addition of dG (comparison between lane 2 and 22, where the latter displayed a more efficient addition); 3) the template containing 8-oxoG enables the incorporation of dG while that containing 8-oxoI displays incorporation of dG with very low efficiencies (lanes 28 and 40); and 4) the corresponding I-containing template enabled the incorporation of dA, while its G-containing RNA analogue does not (in agreement with the result obtained from experiments at lower [RT]) (lanes 37, 25). Furthermore, all experiments carried out in the presence of a mixture containing equal ratios of dCTP/dATP/dGTP/dTTP displayed full cDNA synthesis showing bands at the expected +12N site (Fig 2D, lanes 26, 32, 38, 44). The results indicate that in the case of the canonical template strand (**1**), dCTP keeps adding up to the +4N-position (opposite one additional A and two Cs) stopping at a site where the enzyme encounters a U (lane 23). This is consistent with previous reports where formation of a C:dT (such as in +3N-, and +4N-positions) base pair is likely to form, while a U:dT base pair occurs with very low efficiency [45] (such as at position +5N, where DNA synthesis stalled). The same trend was observed in the case of template strand containing inosine (**2**), albeit with less efficient addition at +3N, and +4N; confirmed by plotting % conversion as a function of time with constant [dCTP] and [AMV RT] (S7 File). On the other hand, reactions carried out in the presence of dCTP with templates containing 8-oxoG (**3**) or 8-oxoI (**4**) only displayed an additional band (+2N - opposite A, S7 File) (Fig 2 - lanes 29, 41). These reactivity changes suggest that anomalies in the structure, due to altered base pair interactions arising from the modification, may affect cDNA synthesis following the incorporation of the first nucleotide.

**dNTP Incorporation in the presence of a canonical/wobble/modified base pair at the end of RNA:DNA duplex (1:6–4:6 / 1:7–4:7 / 1:8–4:8).**   To probe for cDNA synthesis in the presence of a modification already in place, we extended the length of the primer and carried out reactions using 18-mers of DNA **6** (with a 3'-terminal dA), **7** (with a 3'-terminal dC), or **8** (with a 3'-terminal dT) (Fig 3A). Thermal denaturation transitions of duplexes **1:6–4:6** displayed some slight differences, with duplex containing an 8-oxoI:dA base pair (**4:6**) yielding lower values ($\Delta T_m \approx 0.2°C$); and overall higher than those observed with analogous duplexes **1:5–4:5** ($\Delta T_m \approx + 0.2$–$1.1°C$), due to the addition of a base pair interaction. Furthermore, $T_m$ values corresponding to duplexes **1:7–4:7** (containing an additional dC) displayed 1) a clear increase in stability arising from the formation of a Watson-Crick (WC) base pair in **1:7**; 2) destabilization in the presence of an 8-oxopurine:dC base pair compared to its 8-oxopurine:dA analogue (**3:6** and **4:6** compared to **3:7** and **4:7** respectively); and 3) values that were equivalent for an I:dC or I:dA basepair in duplexes **2:6** and **2:7**. Thus indicating that 8-oxopurines form more stable base pairs with dA than dC at the ends of an RNA:DNA duplex; a result that is in agreement with reverse transcription experiments, where templates containing I are able to facilitate the incorporation of dA. Interestingly, RNA:DNA duplexes containing a dT at the end (**1:8–4:8**) showed equivalent values.

**(A)**

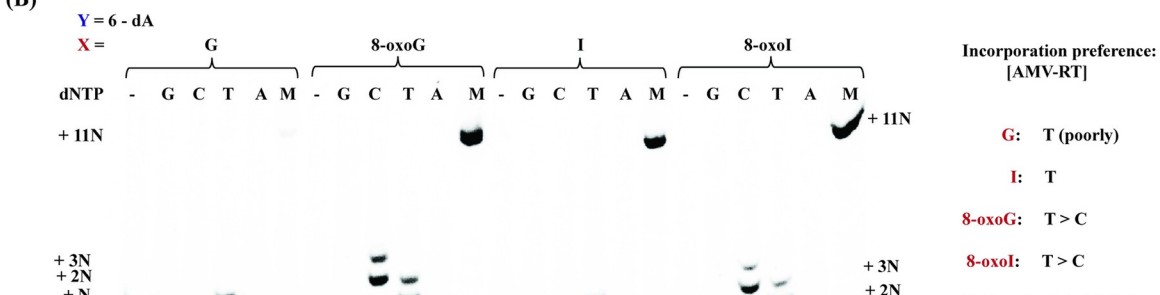

1 - 4    3'- CUC CAC ACA UCC ACC AC**X** ACC UCA ACU GU
5' - HOP*(O)₃ - GAG GTG TGT AGG TGG TG**Y**

**X** = 1 - G; 2 - I; 3 - 8-oxoG; 4 - 8-oxoI       **Y** = 6 - dA; 7 - dC; 8 - dT

| Duplex | $T_m$ (°C) | Duplex | $T_m$ (°C) | Duplex | $T_m$ (°C) |
|---|---|---|---|---|---|
| 1:6 | 71.9 ± 0.4 | 1:7 | 73.4 ± 0.8 | 1:8 | 72.4 ± 0.4 |
| 2:6 | 72 ± 0.2 | 2:7 | 71.7 ± 0.4 | 2:8 | 71.7 ± 0.6 |
| 3:6 | 72.3 ± 0.6 | 3:7 | 70.6 ± 0.6 | 3:8 | 72.2 ± 0.5 |
| 4:6 | 71.1 ± 0.4 | 4:7 | 70.8 ± 0.3 | 4:8 | 71.5 ± 0.5 |

**(B)**

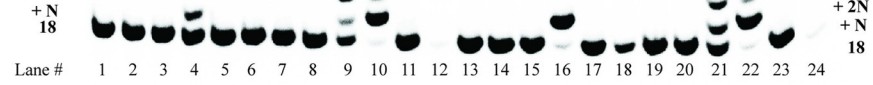

Y = 6 - dA

Incorporation preference: [AMV-RT]

G:   T (poorly)

I:   T

8-oxoG:   T > C

8-oxoI:   T > C

Full synthesis inhibited on canonical RNA-template

**(C)**

Y = 7 - dC

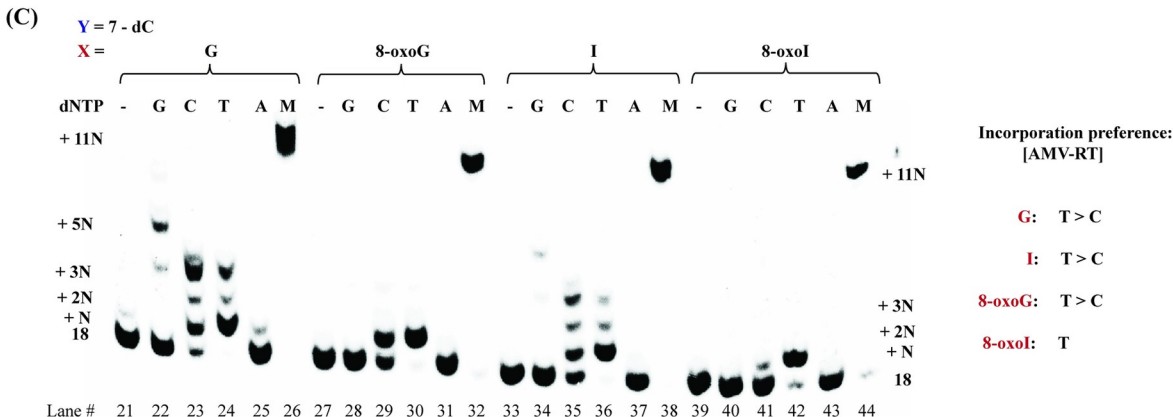

Incorporation preference: [AMV-RT]

G:   T > C

I:   T > C

8-oxoG:   T > C

8-oxoI:   T

**(D)**

Y = 8 - dT

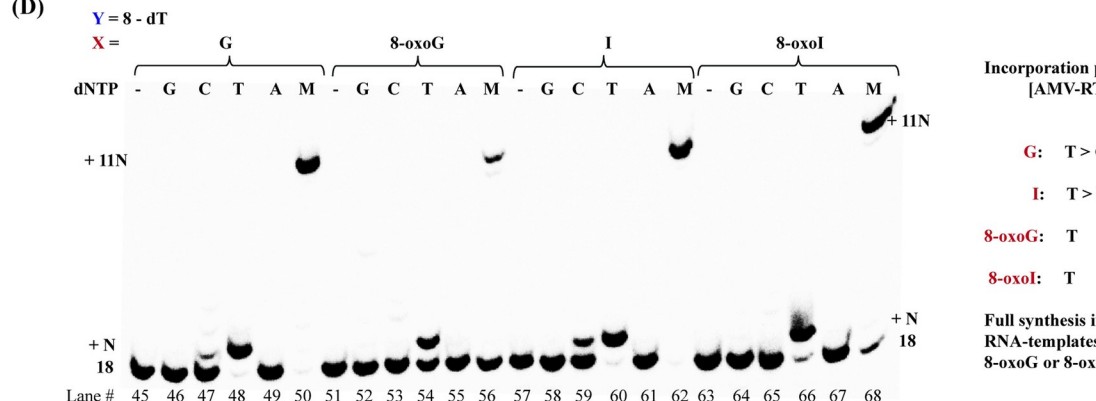

Incorporation preference: [AMV-RT]

G:   T > C

I:   T > C

8-oxoG:   T

8-oxoI:   T

Full synthesis inhibited on RNA-templates containing 8-oxoG or 8-oxoI

**(E)**

| X | dNTP | ON 6:X | | | | ON 7:X | | | | ON 8:X | | | |
|---|---|---|---|---|---|---|---|---|---|---|---|---|---|
| | | $V_{max}$ | $K_m$ | $V_{max}/K_m$ | $K_{cat}$ | $V_{max}$ | $K_m$ | $V_{max}/K_m$ | $K_{cat}$ | $V_{max}$ | $K_m$ | $V_{max}/K_m$ | $K_{cat}$ |
| | | (%·min⁻¹) | (µM) | (%·min⁻¹·µM⁻¹) | (min⁻¹) | (%·min⁻¹) | (µM) | (%·min⁻¹·µM⁻¹) | (min⁻¹) | (%·min⁻¹) | (µM) | (%·min⁻¹·µM⁻¹) | (min⁻¹) |
| G | T | – | – | – | – | 8.6 ± 1.3 | 30.5 ± 4.3 | 0.28 ± 0.2 | 12.3 | 11.3 ± 0.5 | 28.5 ± 19 | 0.4 ± 0.5 | 16.1 |
| 8-oxoG | T | 8.5 ± 0.6 | 0.6 ± 0.3 | 17.3 ± 10.5 | 12.1 | 14.1 ± 2.1 | 11.5 ± 4.3 | 1.2 ± 0.3 | 20.1 | – | – | – | – |
| I | T | 11.2 ± 0.8 | 4.3 ± 4.3 | 5.6 ± 5 | 16 | 7.6 ± 1.1 | 38 ± 10.4 | 0.2 ± 0.2 | 10.9 | 1.03 ± 0.02 | 0.34 ± 0.2 | 3 ± 0.4 | 1.5 |
| 8-oxoI | T | 4.9 ± 0.2 | 193.1 ± 9.8 | 0.03 ± 0.01 | 7 | 16.6 ± 0.6 | 22.3 ± 10 | 0.7 ± 0.2 | 23.7 | 14.1 ± 0.6 | 186 ± 74 | 0.08 ± 0.2 | 20.1 |

**Fig 3.** Thermal denaturation transitions of duplexes **1:6**–**4:6**, **1:7**–**4:7**, and **1:8**–**4:8** (A); and reactivity dNTP incorporation in the presence of AMV RT using duplexes **1:6**–**4:6** (B), **1:7**–**4:7** (C) and **1:8**–**4:8** (D) along with (E) their corresponding steady state kinetics for the most efficient cases. All experiments were carried out under buffered conditions (5 mM Tris-acetate, 7.5 mM Potassium Acetate, 0.8 mM Magnesium Acetate (10 μM free $Mg^{2+}$), 1 mM DTT, pH 8.3, 37 ˚).

Notably, RTn experiments using duplex **1:6** (Fig 3B) did not display incorporation of dTTP efficiently (lane 4) and led to inhibition of cDNA synthesis in the presence of a dNTP mix (evident from the lack of +11N band, lane 6). On the other hand, the use of duplex **2:6** (containing I) displayed specific incorporation of dT and enabled full cDNA synthesis (lanes 16, 18), again indicating that I forms a seemingly stable interaction with dA that enables cDNA synthesis in this sequence context. Furthermore, the use of templates containing either oxidative lesion, 8-oxoG or 8-oxoI (duplexes **3:6** and **4:6**), enabled the incorporation of dT as well as cDNA synthesis (lanes 10, 22, 12, 24). The presence of these oxidized nucleotides also allowed the incorporation of dC, albeit less efficiently, evidenced by the presence of a band corresponding to unreacted DNA primer **6** (lanes 9, 21). The fact that full cDNA synthesis follows the efficiency trend, 8-oxoG ≈ 8-oxoI > I (lanes 12, 18, 24), also points to significant differences in reactivity between templates containing either oxidative lesion and those containing I, with inhibition in the presence of a canonical G:dA base pair in duplex **1:6**.

To corroborate the dependence of reverse transcription with the base pairing interactions at this position, duplexes where the last nucleotide of the RNA template is base pairing to dC (duplexes **1:7**–**4:7**) were probed for RTn (Fig 3C). As expected, formation of a Watson-Crick base pair in duplex **1:7** restored the ability of the RT to carry out cDNA synthesis in the presence of a dNTP mix; where the other three duplexes also displayed cDNA synthesis under these conditions (lanes 26, 32, 38, 44). Steady-state kinetics showed that all duplexes enabled the incorporation of dT with similar efficiencies (Fig 3E). In particular, we initially expected dT incorporation to be most efficient upon use of the canonical G:dC-containing duplex (**1:7**), given that this is the canonical analogue of the reaction. It is unclear at the moment on why the templates containing the oxidative modifications, or I, enable the incorporation of dT in this context However, it is possible that this process is facilitated by additional interactions between the RNA:DNA duplex and the RT-enzyme.

To further explore this unexpected reactivity, RTn experiments were carried out using duplexes **1:8**–**4:8**, where the site of interest is base pairing to dT (Fig 3D). In agreement with previous results, the duplex containing an I:dT base pair (**2:8**) displayed the most efficient incorporation of dTTP, over duplexes containing G:dT (7.5 × slower) and 8-oxoI:dT (37.5 × slower), with inefficient dTTP incorporation on duplex **3:8** (containing 8-oxoG). The fact that dTTP incorporation is inefficient in the presence of an 8-oxoG:dT or 8-oxoI:dT base pair is demonstrated in experiments carried out in the presence of a mix of all dNTPs, where cDNA synthesis is halted or partially inhibited (lanes 50, 56, 62, 68).

**dNTP Incorporation using RNA:DNA duplexes containing 8-bromoinosine.** To gain more insight into potential H-bonding effects arising from I and 8-oxoI, we prepared oligonucleotides containing 8-BrI. It is well established that placing a substituent at the C-8 position changes the equilibrium in favor of the isomer with a *syn*-conformation, thus potentially changing its H-bonding pattern (Fig 4, left). Comparing results between I (anti-) and/or 8-oxoI (syn-) enables one to learn about potential H-bonding interactions arising from the N-7 and C-8 positions, as well as conformational changes, and how these may affect interactions with the enzyme. Rotation around the glycosidic bond may aid in establishing differences between 8-BrI and I, as well as changes in H-bonding patterns (both 8-BrI and 8-oxoI are expected to exist, preferentially, in their syn-conformation) between 8-BrI and 8-oxoI (Fig 4, left). To this end, the RNA template containing 8-BrI (**9**) was prepared and cDNA synthesis was explored in the presence of DNA primers **5**–**8** using AMV RT. As indicated within Fig 4

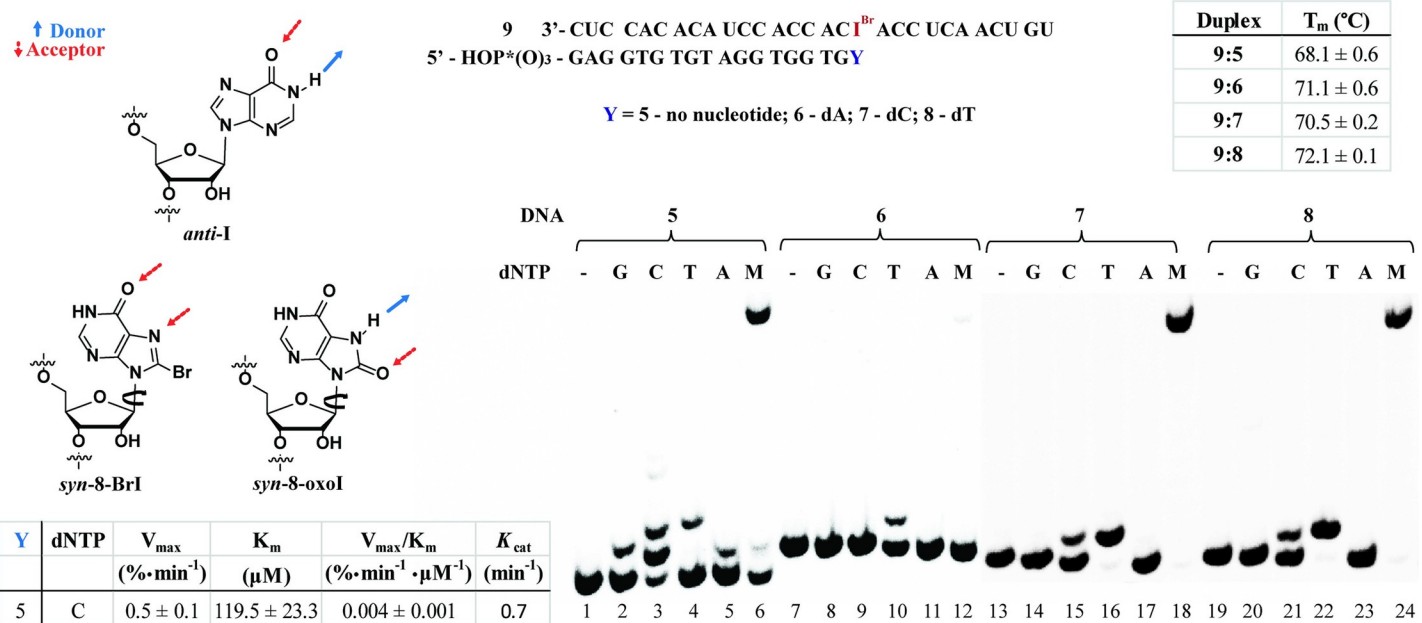

**Fig 4.** Thermal denaturation transitions of duplexes **9:5**–**9:8** and H-bonding patterns expected from I, 8-oxoI, and 8-BrI (left); and RTn using AMV on duplexes **9:5**–**9:8** in the presence of canonical dNTPs, where M = equimolar mixture of all dNTPs, (right). Steady-state kinetics data is shown in the lower left corner. All experiments were carried out under buffered conditions (5 mM Tris-acetate, 7.5 mM Potassium Acetate, 0.8 mM Magnesium Acetate (10 µM free $Mg^{2+}$), 1 mM DTT, pH 8.3, 37 ˚).

(upper-right insert), we were surprised to find that this chemical modification induces thermal destabilization on duplex **9:5**, compared to analogue **2:5** (Fig 2A), suggesting that it is somehow interacting to destabilize this duplex in the absence of a direct base pair opposite 8-BrI. On the other hand, a duplex containing an 8-BrI:dT base pair (**9:8**) resulted in the more stable species, with duplexes **9:6** and **9:7** displaying equivalent $T_m$ values; which suggests that this chemical modification base pairs with similar strengths to dC or dA (at the end of the duplex). RTn using AMV RT and duplex **9:5** showed that the incorporation of dNTPs opposite 8-BrI followed the trend: C > T ≈ A ≈ G, with inefficient cDNA synthesis in the presence of dNTP mix (lanes 1–6). Steady-state kinetics showed that the incorporation of dCTP opposite 8-BrI is about 5 orders of magnitude less efficient than the analogous process for an RNA template containing I (Fig 2C). This result suggests that 8-BrI is in its anti-conformation, as the pattern is similar to that observed when using RNA template **2**, containing I (Fig 2A), and that the bromine at the C8-position adversely affects cDNA extension. In addition, RTn showed that duplexes formed using templates **7**, and **8** led to incorporation of dT as well as cDNA synthesis in the presence of dNTP mix (lanes 16, 22, 18, 24), while hybridization with DNA primer **6** (containing an 8-BrI:dA base pair) did not show a band corresponding to this N+11 product (lane 12). The results indicate that the 8-BrI:A base pair does not form a stable interaction, which differs from the result obtained using an I:A base pair and suggests that bromine induces an adverse interaction in this case. On the other hand, the observation that duplex **9:7** enables the incorporation of dT and restores cDNA synthesis, suggests that the 8-BrI:C base pair is stable enough to allow for the incorporation of a dNTP (in this case dT) and subsequent DNA synthesis. Importantly, there is no direct relationship between thermal stability of the RNA:DNA duplex and RTn efficiency in these cases, thus suggesting that enzyme:duplex:dNTP interactions may differ upon cDNA synthesis.

**dNTP Incorporation RNA:DNA duplexes containing a canonical or modified base pair two positions away from RTn initiation (10 / 11).** Given the observed differences in cDNA

synthesis between RNA templates containing G/I and 8-oxoG/8-oxoI, e.g., addition of dCTP in the **1:5**–**4:5** family (Fig 2B), we decided to probe the effect of a potential modified base pair, 2-positions away from the site of interest, arising from the corresponding modifications. We explored RTn using DNA 19-mers **10** and **11** as models since there was a particular interest in observing the ability of the oxidatively generated lesions to base pair with dC or dA (Fig 5, top). To understand potential changes arising from destabilization of H-bonding interactions, the $T_m$ values were obtained (Fig 5, inset). Noteworthy observations are that: 1) the canonical duplex **1:11** exhibited the highest stabilization (expected since this is the WC base pair); 2) Inosine exhibited slightly higher stability upon base pairing to dC over dA; 3) 8-oxoG displayed no difference between base pairing to dC or dA, and the value was close to that observed with an I:A base pair; and 4) 8-BrI base pairs with preference for dA over dC, and generally with lower values than those observed with the other modifications/lesions. We then carried out the corresponding experiments using AMV RT and observe that only dG was incorporated opposite C (on RNA template), in agreement with results from the other duplexes where dG did not add efficiently opposite any canonical, or modified, nucleobase. Interestingly the % ratio of dG incorporation at the N+2 position was similar in all cases, however, reactions carried out in the presence of dNTP mixtures (Fig 5, columns labeled M) displayed differences in overall cDNA synthesis efficiency. Side-by-side reactions carried out under the same conditions showed that the efficiency for reverse transcription to an N+10 site was generally enhanced upon using the DNA containing a dC, with the following trend: G:C > G:A ≈ 8-oxoG: C > 8-oxoG:A ≈ I:C > I:A >> 8-oxoI:C ≈ 8-oxoI:A, where the corresponding base pair corresponds to position 18. Furthermore, the use of templates containing 8-BrI in duplexes **9:10** or **9:11** led to incorporation of dG and full cDNA synthesis, where the duplex containing an 8-BrI:A base pair incorporated dG more efficiently (S8 File).

**Incorporation of dATP opposite I is pH dependent.** The unexpected reverse transcriptase activity on templates containing I, i.e., that RNA template **2:5** facilitates the incorporation of dATP on the DNA primer in the presence of AMV RT; or that contrary to duplex **1:6**,

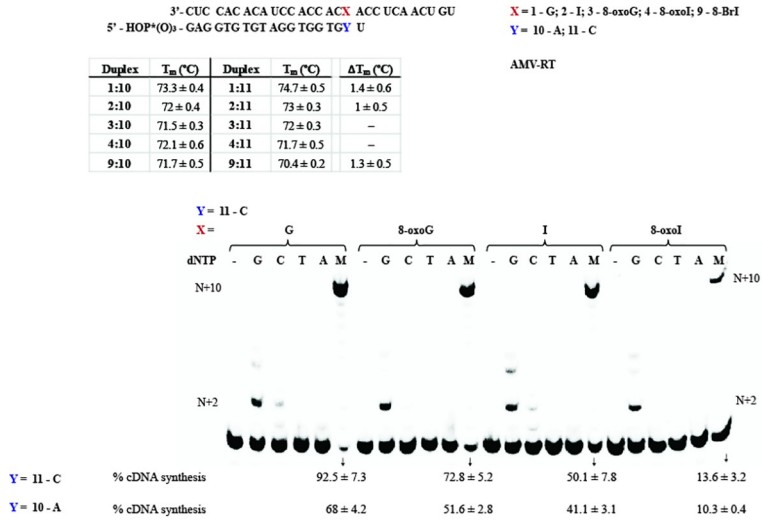

**Fig 5. Thermal denaturation transitions of duplexes 1:10–4:10 and 9:10 as well as 1:11–4:11 and 9:11 along with the corresponding RTn experiment using AMV RT.** The gel corresponding to **1:11**–**4:11** and **9:11** is included in the S9 File, where only the ratios in cDNA synthesis varied. $\Delta T_m$ corresponds to the difference between RNA:DNA hybrids containing **10** and **11**. All RTn experiments were carried out under buffered conditions (5 mM Tris-acetate, 7.5 mM Potassium Acetate, 0.8 mM Magnesium Acetate (10 μM free $Mg^{2+}$), 1 mM DTT, pH 8.3, 37 ˚).

duplex **2:6** enables cDNA synthesis; motivated us to explore this process in more detail. We hypothesized that this reactivity was due to the formation of seemingly stable interactions between I and dA, and that varying the conditions could lead to changes in selectivity. Since all of the buffers provided by the manufacturer are at a pH value of 8.3, we decided to prepare buffered solutions with the same salt concentrations and different pH values of 5.5, 7.3, 8.4 (as control), or 9.5. Importantly, no bands corresponding to degradation/cleavage products were observed upon subjecting the oligonucleotides to this pH range under the described experimental conditions. Control reactions using duplexes **1:5**–**4:5** at pH 8.4 displayed results that were equivalent to those described on Fig 2A; and increasing the pH to 9.5 led to a drastic decrease in selectivity in the case of every RNA template. On the other hand, decreasing the pH closer to physiological conditions of 7.3 resulted in inhibited addition of dATP opposite I, while displaying the same reactivity on templates containing G, 8-oxoG, or 8-oxoI. Experiments carried out at different incubation times, as low as 5 min, displayed the same trend. Thus suggesting that the formation of I:dA interactions may not be biologically relevant, or dependent on H-bond interactions that are facilitated by pH values greater than physiological values. Furthermore, lowering the pH to 5.5 decreased the activity of the reverse transcriptase in a significant manner. The set of data for these experiments is provided within S10 File.

**Reverse transcription using MMLV RT.**   To explore potential differences/similarities of reactivity amongst reverse transcriptases, we probed for differences in fidelity and reactivity using MMLV RT. As illustrated in S11 File, the selectivity and reactivity was similar to that described for AMV RT, with the only measurable difference in the incorporation ratios of dC or dA opposite 8-oxoI or 8-oxoG. However these were only observed at high enzyme concentrations, thus both RTs have overall similar selectivity against the canonical dNTPs. In addition, experiments carried out using DNAs **6** or **7** as primers led to similar results where the use of DNA **6** led to cDNA synthesis inhibition in the case where a G:A base pair is present at the end while the rest of the modifications (I/8-oxoG/8-oxoI) allowed for full cDNA synthesis. As expected, cDNA synthesis is restored upon using DNA **7** with a canonical G:C base pair at this position (full cDNA synthesis is also observed in all other cases). Furthermore there was no difference in the trends of dNTP addition upon using Superscript-II as RT (S12 File), which is not surprising given that this RT is derived from a MMLV source.

**Reverse transcription using HIV RT.**   Taking into consideration that the HIV RT is a less selective/specific reverse transcriptase, we set out to probe its reactivity towards the modifications in this work. As shown in Fig 6, the use of duplexes **1:5**–**4:5** showed that dGTP was incorporated opposite the site of interest with low efficiencies (lane 2), unless higher [HIV RT] were used (S13 File); and all purine nucleobases added dC, dT, or dA in varying ratios. The selectivity between G- and I-containing templates (incorporating the dNTP opposite the site of interest), showed that the G-containing template: (**1**) enables incorporation of its WC-base pair dC as well as dT (lanes 3,4) while the I-containing analogue; and (**2**) yields incorporation of dT, dA, and dC with similar efficiency (based on product ratios). Furthermore, contrary to the other RTs, HIV RT facilitated the incorporation of dA opposite G, albeit less efficiently than its analogous I-containing template (**2:5**, lanes 5 and 16). The trends between duplexes **3:5** (8-oxoG) and **4:5** (8-oxoI) were very similar in the presence of dCTP or dATP (lanes 9, 11, 19, 21), however the latter led to insertion of dTTP more efficiently than the 8-oxoG template (lanes 10 and 20). Which highlights the effect arising from the lack of an exocyclic amine and its role allowing for interaction between inosine derivatives with dT. Strikingly side-by-side comparison between duplexes **1:6**–**4:6** and **1:7**–**4:7** (Fig 7) showed that full cDNA extension (in the presence of dNTP mix) was inhibited, or heavily affected (lanes 12, 24, 36, 48), in every case where an A was base pairing with the site of interest. On the other hand, efficient cDNA synthesis proceeded in the case where a canonical G:C base pair was formed (duplex **1:7**, lane

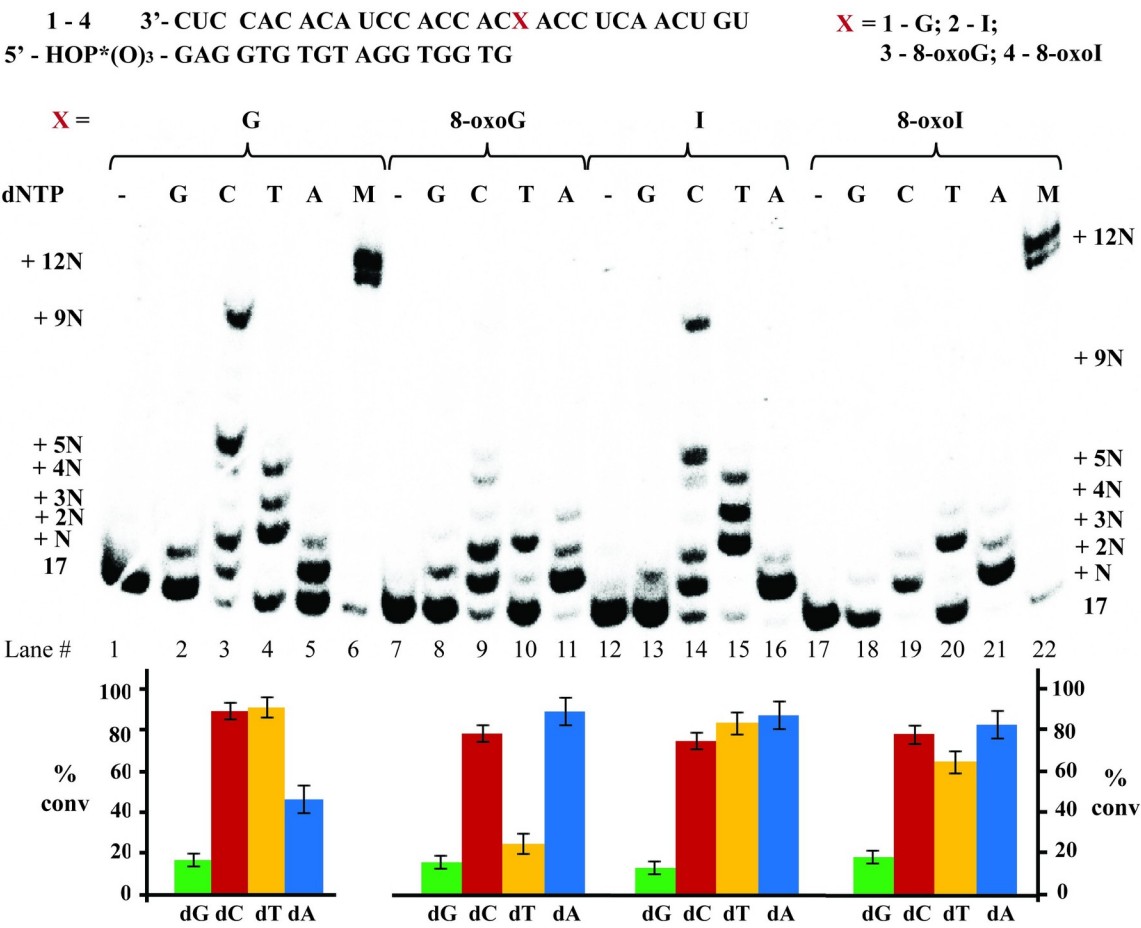

**Fig 6. Reverse transcription using HIV RT with RNA:DNA duplexes 1:5–4:5 along with the % conversion.** All experiments were carried out under buffered conditions (5 mM Tris-acetate, 7.5 mM Potassium Acetate, 0.8 mM Magnesium Acetate (10 μM free $Mg^{2+}$), 1 mM DTT, pH 8.3, 37 ˚).

6). Experiments carried out on duplexes **1:8–4:8** (Fig 7, bottom—containing a dT opposite the modification) also led to efficient addition of dTTP (and cDNA synthesis) when using templates containing G, I, or 8-oxoI (52, 54, 64, 66, 70, 72), with 8-oxoG as the exception (lanes 58, 60). Lastly to explore the impact of a modified base pair two positions away from the transcription site, experiments were carried out using duplexes **1:10–4:10** and **1:11–4:11**. Interestingly, the presence of a modified base pair had a big impact and efficient cDNA elongation was only observed in the cases where a G:dC, I:dC, or G:dA (**1:11**, **2:11**, **1:10**) base pair was present. Suggesting that reverse transcription stalls upon encountering an oxidative lesion in this context. Supporting this observation is the fact that duplexes containing 8-BrI (**9:10 / 9:11**) also resulted in an inefficient process (see S14 File for this experiment series).

## Discussion

The ability of three reverse transcriptases to enable the synthesis of cDNA past I, 8-oxoG, 8-oxoI, or 8-BrI, was examined by focusing on position 18 of a 29-nt long RNA template. The use of AMV RT led to similar results to those obtained with MMLV RT, or HIV RT (slightly varying outcomes observed).

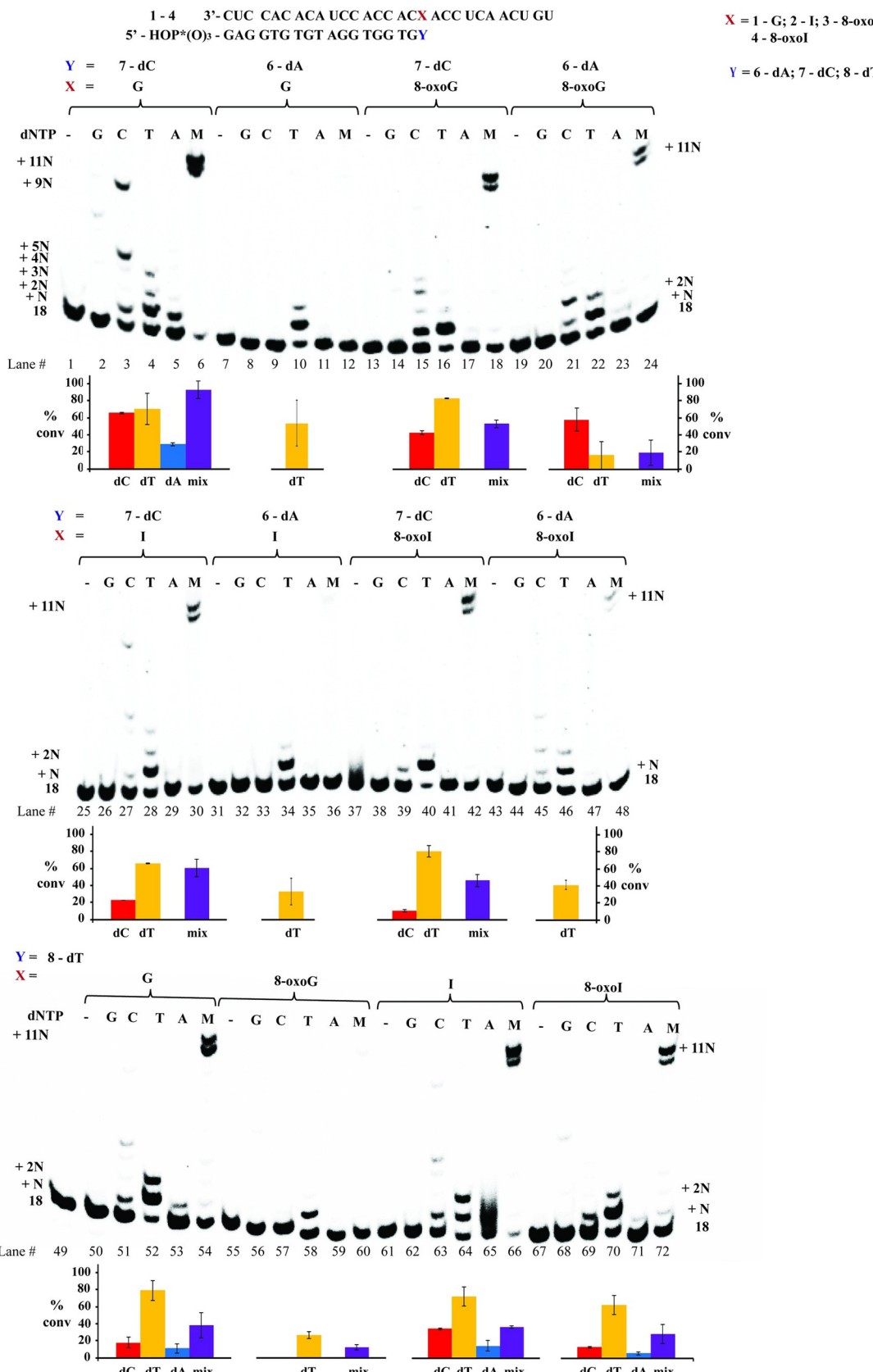

**Fig 7. Reverse transcription using HIV RT with RNA:DNA duplexes 1:6–4:6 / 1:7–4:7 / 1:8–4:8, along with % conversions (shown underneath each gel).** All experiments were carried out under buffered conditions (5 mM Tris-acetate, 7.5 mM Potassium Acetate, 0.8 mM Magnesium Acetate (10 μM free $Mg^{2+}$), 1 mM DTT, pH 8.3, 37 ˚).

## Reverse transcription using RNA templates containing inosine

**I:dC.** The reactivity where RNA templates containing G or I can efficiently incorporate dC was expected [44], and can be explained via formation of canonical WC-type interactions. The incorporation of dC opposite RNA templates containing G was measured as a more efficient process (app. 1.6 ×) than the corresponding I-containing analogue. An observation that can be explained by the lack of a H-bond interaction in I, compared to that expected from the exocyclic amine in G.

**I:dT.** Interestingly templates containing G or I enabled the incorporation of dT albeit with less efficient rates, compared to the analogous addition of dC, of app. 930× and 390× for G or I respectively (comparison amongst duplexes **1:5** or **2:5**, Fig 2). This is in agreement with results obtained on the duplex family **1:8–4:8** where the DNA:RNA duplexes containing an I:dT or G:dT base pair enabled cDNA synthesis, with 8-oxoI:dT following in efficiency (Fig 3D). Where, steady-state kinetics showed that sample **2:8** (containing I:dT) allowed for dTTP incorporation app. 7.5× faster than the canonical analogue **1:8**, and 37.5× more efficiently than duplex **4:8** (containing 8-oxoI:dT). Stable base pair interactions have been previously reported on G:U wobble pairs and can be rationalized, in part, from the formation of stable G:T / I:T H-bonds that are independent of the exocyclic amine (Fig 8A); thus presumably facilitating dTTP incorporation and/or cDNA elongation. Another aspect to consider, particularly in cases where the modification is already in place (duplexes **1:8–4:8**) involves the role of electrostatics in the elongation rate, as determined from crystal structures bound to DNA duplexes [46,47]. Taking into consideration that the presence of G:U wobble pairs induce structural differences to form an electronegative environment in the major groove [48], it is reasonable to expect that this will also have an impact on RT─RNA binding and/or processivity. While this relationship was initially surprising, G:U and I:U wobble pairs have also shown similar reactivity in other enzymatic contexts, such as in translation elongation [49].

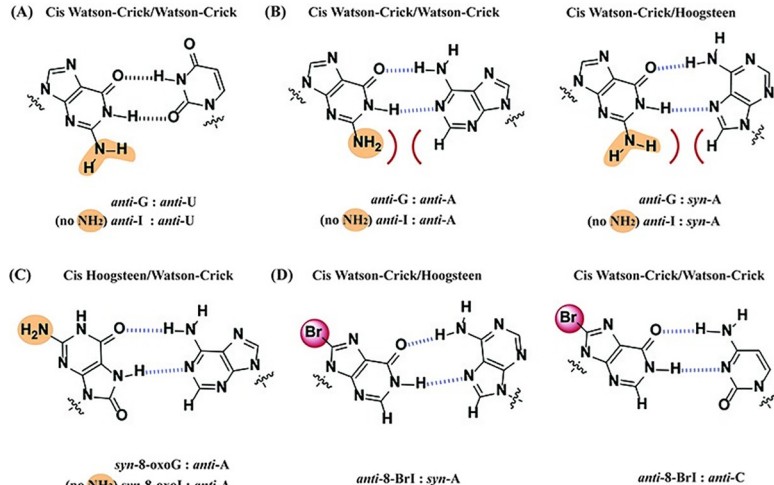

**Fig 8.** Wobble base pairs of G:U (A); of G:A displaying potential steric hindrance with the exocyclic amine, which is not present on I:A base pairs (B); base pairing between 8-oxoG:A, which places the exocyclic amine away from the H-bonding face (C); and possible base pair of 8-bromoinosine with A and C (D).

**I:dA.**    Contrary to the reactivity observed with the use of RNA templates containing G at the site of interest, templates containing I also enabled the incorporation of dA in the presence of the RTs used in this work, albeit in a much less efficient manner (Fig 2B, lane 15). In this regard, G:A base pairs have been reported [50] and base pairing involving I and dA has been characterized, via crystallography, in the context of the ribosomal decoding center [51]. An interaction that may be enhanced due to the lack of the C2-exocyclic amine (Fig 8B). The same behavior was corroborated in experiments where duplexes **1:6**–**4:6** were used, and contrary to 8-oxoG/I/8-oxoI (which allowed cDNA synthesis), duplex **1:6** (containing a G:A base pair at the start site) did not display incorporation of any dNTP efficiently and resulted in cDNA synthesis inhibition (Fig 3B, lanes 1–6). Furthermore, steady state kinetics on this family of duplexes displayed that the trend in efficiency for reverse transcription that occurred when encountering inosine base pairing with a deoxynucleotide at the start site was: I:dT > I:dC > I:dA (Fig 3D, 3C and 3B respectively). Although this is an inefficient process we were interested in probing its prevalence as a function of pH, where lowering the pH to 7.3 abolished the addition of dA opposite I. Since it is unlikely that the protonation states of the nucleobases change in this range, the most probable cause for this observation may be due to varying interactions between the enzyme:duplex:dNTP complex. Importantly this trend has been previously reported, where decreasing the pH leads to increased fidelity while increasing the pH leads to the opposite result [52].

**Reverse transcription using templates containing 8-oxoG or 8-oxoI.**    In agreement with previous reports [38], the RNA template containing 8-oxoG (**3**) allowed for incorporation of dA with a slight preference over dC (app. 1.5× more efficient). It can be assumed that the expected H-bonding interactions between 8-oxopurines, in its syn-conformation, and A play a role in this behavior (Fig 8C). This is also in agreement with 8-oxoG exhibiting H-bonding interactions as Uridine, arising from a conformational change around the glycosidic bond, in other enzymatic contexts [53]. Interestingly, the corresponding 8-oxoI analogue also allowed for incorporation of dC and dA, with higher efficiency in the presence of dATP (app. 20×, Fig 2B). The fact that the RNA template containing I enabled the incorporation of dA more efficiently than dC (4.3× - **2:5** compared to **1:5**) suggests that the C2- exocyclic amine may be playing a significant role within the active site. This pattern was also observed in duplexes that already contained an 8-oxopurine base pairing to dC or dA (**1:6**–**4:6**, **1:7**–**4:7**); where the use of DNA templates containing an additional dA did not allow for efficient cDNA synthesis on the canonical duplex **1:6**. The results highlight the ability of both 8-oxopurines, as well as inosine, to form catalytically active base pairs in this context. Unexpectedly, the use of duplexes **1:7**–**4:7** (containing an additional dC) led to similar rates of dT incorporation, with the trend: 8-oxoG > 8-oxoI > G ≈ I. This result suggests that while the formation of stable duplexes at the end is necessary, other RNA-protein interactions may have an impact in the final outcome of the biochemical process. It is possible that the Hoogsteen-face of the 8-oxopurines facilitates interactions between the enzyme-RNA:DNA-nucleotide components to facilitate addition of dT in this case. Binding affinities of AMV RT to different duplexes have been established as RNA:DNA >> RNA ≈ DNA:DNA > DNA [54] and may vary depending on the RT in use, pointing to the relationship between differences in secondary structure and the ability of the RT to carry out its functions. Lastly, the use of duplexes **1:8**–**4:8** (containing an additional dT) led to incorporation of dTTP for 8-oxoI and not for 8-oxoG (S15 File), which suggests that 8-oxoI may be behaving conformationally, like inosine in its anti-conformation.

**Reactivity on templates containing 8-bromoinosine.**    To learn more about the potential H-bonding interactions, and understand more on the role of a modification at the C8-position, the phosphoramidite of 8-BrI was used to prepare RNA templates with this moiety. Thermal denaturation transitions showed that 8-BrI can form stable base pairs with the trend dT > dA

> dC (Fig 4), and RTn experiments carried out on duplex **9:5** led to incorporation of dC preferentially. Indicating that the formation of a stable base pair with the incoming dNTP is not an accurate indicator of what will occur when the modification is already in place (duplexes **9:6**–**9:8**). Furthermore, it was found that RNA modified with an 8-BrI at the start site facilitates the insertion of dC or dT opposite to it and not dA, suggesting that the I:dA base pair likely forms with I in the anti-conformation. Thus 8-BrI may potentially base pair with both dC and dA as proposed on Fig 8D. The fact that templates containing 8-BrI do not lead to cDNA synthesis, while those containing I do, implies that the bromine may be posing adverse interactions that may prevent protein:RNA contacts that affect transcription efficiency. Pointing out to the importance of having these interactions intact and that the conformation and H-bonding between base pairs is not sufficient to dictate enzymatic reactivity, since other aspects and changes in structure must be taken in consideration [55].

**Reactivity using HIV RT.** The results obtained with duplexes **1:5**–**4:5** in the presence of HIV RT were generally similar with differences that may be due to experimental conditions, e.g., this was the only RT that enabled incorporation of dA opposite an RNA template containing G but less efficiently than the corresponding I-analogue. Base pairing involving I:A can be rationalized from the formation of anti-I:anti-A or anti-I:syn-A (Fig 8B), with the latter previously reported [56], thus making this a more likely candidate. In addition the thermal stability trend for base pairing for oligonucleotides of DNA containing dI is I:C > I:A > I:T ≈ I:G [57], also supporting the formation of a relatively stable base pair. Another difference was that templates containing 8-oxoI had no selectivity between the incorporation of dA or dC, contrary to experiments in the presence of AMV RT, where duplex **4:5** exhibited more efficient incorporation of dATP. It is important to note that kinetic data is still needed to confirm these trends. On the other hand, there was no selectivity towards insertion of dA or dC in the presence of templates containing 8-oxoG (**3:5**). This result is in contrast with previous reports that used DNA duplexes containing 8-oxoG, where HIV RT incorporated dCTP more efficiently than dATP [58]. This may be due to differences in substrate (RNA vs DNA) as well as experimental conditions. Another difference can be observed upon comparison of duplexes containing 8-oxoI and 8-oxoG, where experiments using the former led to incorporation of dT in higher conversion ratios. Which confirms the importance, and impact, that the exocyclic amine may have in this and other enzymatic contexts. Overall, the use of duplex **2:5** (containing inosine) suggests that this modification can code like G, A, and U to a lesser extent and that this process is pH dependent. The use of DNA 18-mers **1:6**–**4:6** and **1:7**–**4:7** shows that the HIV RT is effective at continuing cDNA synthesis if there is a canonical base pair at the 3'-end of the DNA, while both AMV and MMLV are less affected by the presence of a modification at this position. The same behavior was also observed when an oxidative lesion is two positions away from the start of RTn.

Overall, the results obtained using 8-oxoG and AMV or MMLV RT are in good agreement with a previous report [38], where A and C added efficiently opposite 8-oxoG compared to addition of C and T opposite G. In a separate report, C and T were reported to extend DNA synthesis opposite 8-oxoG using RAV2 RT and exclusively C using HIV RT [59]. This report contrasts with the results obtained herein, possibly due to the use of different experimental conditions. Another important aspect to take into consideration regards to the relationship between concentration of divalent ions (in this case Magnesium) and RT fidelity, where Mg concentration can impact the outcome depending on the biological system, where lower fidelity occurs with HIV RT and not with MMLV RT [60]; or may alter the overall translation mechanism in other systems [61]. The work described herein was carried out in free $Mg^{2+}$ concentrations that ranged from 10–370 μM range, which approaches physiological ranges of some biological systems (although the concentration of dNTP used is much higher than what

would be encountered *in vivo*) and is on the lower end of concentrations typically used *in vitro*. Another aspect of consideration is the dNTP concentration used in this work, where increasing the concentration of dNTP led to transcription inhibition presumably due to a large amount of dNTPs chelating to the divalent cation. Interestingly, dNTP concentrations up to 20 mM have been effectively used (while using the concentrations for all other species reported in this work) with other RNA modifications and the three reverse transcriptases used [62], which suggests that the concentration of Mg at trace levels is sufficient to carry out the desired reactions *in vitro*.

## Conclusion

The reactivity described provides important information on the behavior of oxidatively generated lesions, or inosine, within RNA and their role on reverse transcription. A reliable method for the synthesis for the phosphoramidites of 8-oxoinosine and 8-bromoinosine along with their subsequent incorporation into RNA is reported. Importantly, the findings described herein shed light onto this process in cases where: 1) the process starts at the site of modification; 2) the modification/lesion is already involved in a base pair interaction; or 3) the modification/lesion is two bases away from the start of RTn. It is important to highlight that when analyzing some of the processes reported herein, e.g., I:dT or I:dA interactions, one must take into consideration the kinetics data to assess on the likelihood that they may be relevant in vivo (or within the active site under varying local H-bonding networks). Furthermore, the dependence on pH that was observed suggests that while I seems to be decoded as A or U (in addition to the more commonly accepted G),[7] this process may not be relevant in vivo. The use of inosine and 8-oxoinosine pointed to the impact that the exocyclic amine has on reverse transcription recognition and efficiency. Since a link between the presence of inosine, and/or oxidative stress, on viral pathogenesis exists, the information provided herein will be useful. It is plausible that reverse transcriptases encounter the lesions/modification explored in this work, thus a good understanding on how RNA templates containing them will be useful in assessing their impact.

## Supporting information

**S1 File. Experimental details for the synthesis of the phosphoramidite of 8-oxoI.** Includes experimental details; $^1H$ and $^{13}C$ NMR spectra of S1; FTIR of S1; $^1H$ and $^{13}C$ NMR spectra of S2; FTIR of S2; $^1H$ and $^{13}C$ NMR spectra of S3; FTIR of S3; $^1H$ and $^{31}P$ NMR spectra of S4.
(PDF)

**S2 File. Experimental details for the synthesis of the phosphoramidite of 8-BrI.** Includes experimental details; $^1H$ and $^{13}C$ NMR spectra of S5; FTIR of S5; $^1H$ and $^{13}C$ NMR spectra of S6; FTIR of S6; $^1H$ and $^{13}C$ NMR spectra of S7; FTIR of S7; $^1H$ and $^{31}P$ NMR spectra of S8.
(PDF)

**S3 File. MALDI-TOF of oligonucleotides 1–4 and 9.** Details and mass spectra of oligonucleotides 1–4 and 9.
(PDF)

**S4 File. UV-vis spectra of HIV, MMLV, and AMV RTs.**
(PDF)

**S5 File. CD and $T_m$ analyses.** Typical CD of RNA/DNA duplex at 20°C and 85° C; CD and $T_m$ measurement for duplex **1:5**; CD and $T_m$ measurement for duplexes **1:5–4:5; 1:6–4:6;** and **9:5, 9:6**; CD and $T_m$ measurement for duplexes **1:7–4:7; 1:10–4:10;** and **9:7, 9:10**; CD and $T_m$ measurement for duplexes **1:11–4:11; 9:11; 1:8–4:8;** and **9:8**; $T_m$ measurements and ANOVA

for duplexes **1:5, 1:6. 1:7, 1:8, 1:10, 1:11**; $T_m$ measurements and ANOVA for duplexes **2:5, 2:6. 2:7, 2:8, 2:10, 2:11;** $T_m$ measurements and ANOVA for duplexes **3:5, 3:6. 3:7, 3:8, 3:10, 3:11;** $T_m$ measurements and ANOVA for duplexes **4:5, 4:6. 4:7, 4:8, 4:10, 4:11;** and $T_m$ measurements and ANOVA for duplexes **9:5, 9:6. 9:7, 9:8, 9:10, 9:11.**
(PDF)

**S6 File. Native PAGE (20%) of RNA w/wo DNA, showing duplex formation in the MMLV buffer (as described in the experimental section).**
(PDF)

**S7 File. Relative rates for 1:5–4:5 with dCTP and 2:5–4:5 with dATP at constant [dNTP] and [AMV-RT] as a function of time.** Reactions carried out at rt.
(PDF)

**S8 File. Duplexes 9:6 and 9:7 in the presence of AMV-RT.**
(PDF)

**S9 File. Duplexes 1:11–4:11 in the presence of AMV-RT.**
(PDF)

**S10 File. Duplexes 1:5–4:5 in the presence of AMV-RT at various pH values.** Top three experiments represent incubation times of 5 min and the bottom experiment represents incubation time of 40 min.
(PDF)

**S11 File.** RNA:DNA 1:5–4:5 at higher (top) and lower (bottom) [MMLV-RT], see table in experimental section for description; and RNA:DNA 1:6–4:6 & 1:7–4:7 at lower [MMLV].
(PDF)

**S12 File. RNA:DNA 1:5–4:5 using SSII (Superscript II).**
(PDF)

**S13 File. Duplexes 1:5–4:5 in the presence of higher [HIV-RT].** HIV units used were ca. 0.187 per well, 30 times more than the experiments described on the manuscript.
(PDF)

**S14 File. Duplexes 1:10–4:10, 1:11–4:11 and 9:10 / 9:11 in the presence of HIV-RT.**
(PDF)

**S15 File. Relative rates for 2:6–4:6 with dTTP and 1:7–4:7 with dTTP at constant [dNTP] and [AMV-RT] as a function of time.** Reactions carried out at rt.
(PDF)

**S16 File.**
(DOCX)

**S17 File. Original uncropped gels.**
(PDF)

**S1 Fig. TOC-1c.**
(TIF)

## Acknowledgments

A.S. would like to acknowledge a Research and Creative Activities Award (RaCAS, CU Denver) for support. Contributions in the synthesis of phosphoramidites and preparation of some

oligonucleotides by Capt. Courtney Kiggins, and Ms. Namoos Siddique are acknowledged. M. K. would like to acknowledge support from Smith College via a summer internship.

## Author Contributions

**Conceptualization:** Marino J. E. Resendiz.

**Data curation:** Marino J. E. Resendiz.

**Formal analysis:** Madeline M. Glennon, Austin Skinner, Marino J. E. Resendiz.

**Funding acquisition:** Marino J. E. Resendiz.

**Investigation:** Madeline M. Glennon, Austin Skinner, Mara Krutsinger, Marino J. E. Resendiz.

**Methodology:** Marino J. E. Resendiz.

**Project administration:** Marino J. E. Resendiz.

**Resources:** Marino J. E. Resendiz.

**Software:** Marino J. E. Resendiz.

**Supervision:** Marino J. E. Resendiz.

**Writing – original draft:** Marino J. E. Resendiz.

**Writing – review & editing:** Marino J. E. Resendiz.

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
