## [Decision Letter · Decision Letter 0]

3 Jul 2020

PONE-D-20-17213

Translesion Synthesis by MmLV-, AMV-, and HIV-Reverse Transcriptases Using RNA Templates Containing Inosine, Guanosine, and Their 8-oxo-7,8-Dihydropurine Derivatives.

PLOS ONE

Dear Dr. Resendiz,

Thank you for submitting your manuscript to PLoS ONE. We have received comments from three experts in the field. All reviewers are aware of the potential interest of your research.  However, they also agree in that the manuscript needs to be improved. Although detailed critiques are appended below this letter, reviewers 1 and 2 have concerns about your assessment of fidelity and the effects of magnesium, while reviewer no. 3 requests accurate estimates of active RT concentration to be used instead of activity units.  The paper needs also some improvement for clarity and readability in its presentation. 

After careful consideration, we think that your manuscript needs MAJOR REVISION in order to be considered for publication. If you are prepared to undertake the work required, we would be pleased to reconsider this decision.

We look forward to receiving your revised manuscript.

Kind regards,

Luis Menéndez-Arias, Ph. D.

Academic Editor

PLOS ONE

Journal Requirements:

2. We note that this submission reports a functional enzymological study with kinetic and thermodynamic data. The reporting of these data should include the temperature, pH and pressure, as well as the identity of the catalyst and its origins, the method of preparation, criteria for purity and assay conditions. We recommend that you refer to the Standards for Reporting Enzymology Data (STRENDA) of the Beilstein Institut for details regarding the adequate description of experimental conditions and reporting of enzyme activity data: https://www.beilstein-strenda-db.org/strenda/public/guidelines.xhtml. Please note that the Beilstein Institut’s STRENDA database automatically checks manuscript data for guideline compliance, as well as making them publicly available after publication and assigning them a specific DOI number for reference and tracking purposes. If you obtain a STRENDA Registry number (SRN) and PDF containing all your functional enzymology data, please include these as Supplementary files.

4. Please include your tables as part of your main manuscript and remove the individual files. Please note that supplementary tables (should remain/ be uploaded) as separate "supporting information" files.

Reviewers' comments:

Reviewer's Responses to Questions

**Comments to the Author**

1. Is the manuscript technically sound, and do the data support the conclusions?

Reviewer #1: Yes

Reviewer #2: Partly

Reviewer #3: Yes

2. Has the statistical analysis been performed appropriately and rigorously? 

Reviewer #1: N/A

Reviewer #2: Yes

Reviewer #3: Yes

3. Have the authors made all data underlying the findings in their manuscript fully available?

Reviewer #1: Yes

Reviewer #2: Yes

Reviewer #3: Yes

4. Is the manuscript presented in an intelligible fashion and written in standard English?

Reviewer #1: Yes

Reviewer #2: No

Reviewer #3: Yes

5. Review Comments to the Author

Reviewer #1: The authors examined the cDNA synthesis reaction in the presence of RMA template containing purine derivatives. The experimental protocol was well-designed and well-conducted. However, the manuscript needs to be improved.

<major points="">

1. According to the title, readers may think this study mainly focus on the comparison of MMLV reverse transcriptase (RT), AMV RT, and HIV-1 RT. However, there is no data for MMLV RT in the main figures. I would like the authors to reconsider if the current title and the order of figures are appropriate.

2. I think tables in the figures should be put together into one or two tables.

3. Resolution of all figures should be increased.

4. Figure legends should be more informative. Enzyme species and concentration, pH, temperature, and reaction time are missing.

5. Generally, the fidelity of reverse transcriptase depends on reaction condition. The error rate increased with increasing MgCl2 concentration. Please discuss possible effects of MgCl2 concentration on the results.

<minor points="">

1. TOC: At a first glance, each color represents dATP, dCTP, dGTP, or dTTP. If it is true, the color of primer is inadequate. Please change colors.

2. Page 5, line 15: “N” should be italicized.

3. Page 10, line 6: I think “(2:5)” should be “(1:5)”.

4. Page 11, line 6: I think “C:dT” should be “U:dC”.

5. Please indicate what “WC” means.

6. Experimental Procedures: The styles of subtitle are different including large capitals of all words or only the first word and italicized or not, which should be uniformed.

7. Figure 2: The difference between Fig. 2B and D is enzyme concentration (0.7 and 2.1 Units, respectively). Almost all substrate remained (lane 2 in Fig. 2B), while no substrate remained (lane 2 in Fig. 2D). Please discuss it.

8. Page 17, line 8 from the bottom: “dC” should be italicized.</minor></major>

Reviewer #2: This manuscript from Resendiz et al. describes the effect of various physiologically relevant G analogs on reverse transcription with AMV, MMLV, Superscript, and HIV reverse transcriptases (RT). The analogs (Inosine, 8-oxo-7,8-dihydroinosine (8oxo-I), 8-oxo-7,8-dihydroguanosine (8-oxoG), and 8-bromo-Inosine) were incorporated into RNA templates (along with a control template with a natural G at the corresponding position) that were primed with DNAs in which the 3’ primer nucleotide was 1 base before the lesion on the template, or at the lesion, or 1 base after. DNA synthesis under all these conditions was examined and quantified. The most extensive work was conducted with AMV RT and the other enzymes were used as comparisons. Overall, there is a lot of useful information in this manuscript and clearly a lot of work was done. Conclusions were justified by statistically relevant data analysis. There were some concerns. The writing was not particularly clear. There were several grammatical errors, too many to list here. Therefore, a PDF document has been provided with several recommendations as comments. Also, and more concerning, the final concentration of free Mg in reactions does not appear to have been considered. There is a general concern that the level of free Mg in reactions may have been very low. With MMLV-RT for example, it is not clear how the reactions worked with 0.3 mM Mg and levels of dNTPs that were above this value. Since dNTPs chelate Mg this would mean the free concentration of Mg in these reactions may have been near 0, essentially trace levels. This needs to be explained. Is it possible that this effected the outcome of some experiments? The reviewers need to reconsider their results after calculating how much free Mg was present. Finally, the fidelity of DNA replication by all the viruses that these RTs were derived from is similar in cells and more recent data indicates that it is also similar when physiological levels of Mg are used in vitro. There are clear differences in fidelity in vitro with high Mg and HIV is less accurate under this condition but is this is not the case with low Mg (see Achuthan et al. 2014, 15:8514). It appears that the results with HIV RT here were mixed and did not necessarily support it being less accurate than the other enzymes. The authors may want to take this into consideration.

Reviewer #3: The manuscript by Glennon and Skinner et al, addresses and interesting biological question regarding DNA synthesis by reverse transcriptases on oxidized bases. The study and manuscript are solid, but a few points remain unaddressed:

1. What is the active site concentration of the enzyme preparations used in this study? This would affect the comparison of different reverse transcriptases, and Table 1 indicates that the polymerases were compared by unit concentration. As the unit assay and definition can vary between different enzymes and different companies, it seems appropriate to measure this for the study and compare the different reverse transcriptases appropriately.

2. Regarding the pH dependence of nucleotide incorporation opposite deoxyinosine: the authors did not address the stability of the RNA template over the pH range tested.

6. PLOS authors have the option to publish the peer review history of their article (what does this mean?). If published, this will include your full peer review and any attached files.

Reviewer #1: **Yes: **Kiyoshi Yasukawa

Reviewer #2: No

Reviewer #3: No

---

## [Author Response · Author response to Decision Letter 0]

22 Jul 2020

The response has been uploaded in the 'response to reviewers' section of the files.

---

## [Decision Letter · Decision Letter 1]

10 Aug 2020

Translesion synthesis by AMV, HIV, and MMLVreverse transcriptases using RNA templates containing inosine, guanosine, and their 8-oxo-7,8-dihydropurine derivatives.

PONE-D-20-17213R1

Dear Dr. Resendiz,

We’re pleased to inform you that your manuscript has been judged scientifically suitable for publication and will be formally accepted for publication once it meets all outstanding technical requirements.

Kind regards,

Luis Menéndez-Arias, Ph. D.

Academic Editor

PLOS ONE

Additional Editor Comments (optional):

Reviewers' comments:

Reviewer's Responses to Questions

**Comments to the Author**

1. If the authors have adequately addressed your comments raised in a previous round of review and you feel that this manuscript is now acceptable for publication, you may indicate that here to bypass the “Comments to the Author” section, enter your conflict of interest statement in the “Confidential to Editor” section, and submit your "Accept" recommendation.

Reviewer #1: All comments have been addressed

Reviewer #2: (No Response)

Reviewer #3: All comments have been addressed

2. Is the manuscript technically sound, and do the data support the conclusions?

Reviewer #1: (No Response)

Reviewer #2: Yes

Reviewer #3: Yes

3. Has the statistical analysis been performed appropriately and rigorously? 

Reviewer #1: (No Response)

Reviewer #2: Yes

Reviewer #3: Yes

4. Have the authors made all data underlying the findings in their manuscript fully available?

Reviewer #1: (No Response)

Reviewer #2: Yes

Reviewer #3: Yes

5. Is the manuscript presented in an intelligible fashion and written in standard English?

Reviewer #1: (No Response)

Reviewer #2: Yes

Reviewer #3: Yes

6. Review Comments to the Author

Reviewer #1: The authors have addressed all the points raised by the reviewers, and in the current form the manuscript can be accepted for publication.

Reviewer #2: The authors carefully considered comments from reviewers and improved this manuscript. They have now included important information about the levels of free Mg and dNTPs in reactions with various enzymes. The fact that several different conditions were used and some reactions had high levels of free Mg while some had essentially tract levels remains a concern. It is not clear why the authors chose this approach and it does make it more difficult to make comparisons. However, since it is now clearly explained, this allows the reader to draw their own conclusions. Even with this issue some of the data in the report will be valuable to others. PLOS one is an appropriate venue for this report.

Reviewer #3: I struggled with this review and revision. The experiments as they were set up are a missed opportunity. The reaction conditions for the kinetic studies were not chosen in a consistent way. The enzyme concentrations were used directly from the manufacturer - as if for providing information for biotechnology use, and vary widely between the enzymes tested (where known)- but the magnesium concentration was chosen to represent in vivo levels (but would not be used in molecular biology applications), yet the dNTP concentration corresponds to what would be used in vitro. However, within these specific conditions, the data and conclusions are valid.

7. PLOS authors have the option to publish the peer review history of their article (what does this mean?). If published, this will include your full peer review and any attached files.

Reviewer #1: **Yes: **Kiyoshi Yasukawa

Reviewer #2: No

Reviewer #3: No

---

## [Editor Report · Acceptance letter]

12 Aug 2020

PONE-D-20-17213R1 

Translesion synthesis by AMV, HIV, and MMLVreverse transcriptases using RNA templates containing inosine, guanosine, and their 8-oxo-7,8-dihydropurine derivatives. 

Dear Dr. Resendiz:

I'm pleased to inform you that your manuscript has been deemed suitable for publication in PLOS ONE. Congratulations! Your manuscript is now with our production department. 

Kind regards, 

on behalf of

Dr. Luis Menéndez-Arias 

Academic Editor

PLOS ONE